# Alterable Electricity Pricing Mechanism Considering the Deviation of Wind Power Prediction

**Qinqin Cai** [1,2]**, Yongqiang Zhu** [1,2,*]**, Xiaohua Yang** [2] **and Lin E** [2]

[1] State Key Laboratory of Operation and Control of Renewable Energy & Storage System, China Electric Power Research Institute Co., Ltd., Beijing,100192, China; 120192201255@ncepu.edu.cn

[2] State Key Laboratory of Alternate Electrical Power System with Renewable Energy Sources, North China Electric Power University, Beijing,102206, China; xiaohuayang@ncepu.edu.cn (X.Y.); 120192201289@ncepu.edu.cn (L.E.)

[*] Correspondence: zyq@ncepu.edu.cn; Tel.: +86-139-1000-2860



**Featured Application: This paper is a policy article that is used to alleviate the problem of power deviation caused by the volatility and uncertainty of wind power. It is also suitable for promoting "source-network coordination" between the power system and the wind farm operator.**

**Abstract:** Fluctuation and prediction errors of wind power would cause a large amount of automatic generation control (AGC) adjustment costs, which lead to the problem of power curtailment. A reasonable mechanism of grid-connection electricity price may encourage wind farms to take measures to reduce the deviation between output power and schedule power, which is helpful for source-network coordination and reducing wind power curtailment. An alterable electricity pricing mechanism considering wind power deviation rate is proposed. In each schedule cycle, electricity price is adjusted according to the deviation rate and its historical change trend. In this way, wind farms will be encouraged to configure energy storage to promote the accordance of wind output power with schedule power to the greatest extent. Given the statistical characteristic of prediction errors of wind power, this paper proposes a schedule power model, taking least squares of output power deviation as objective function, and then puts forward an engineering application method for determining schedule power. This paper analyzes the overall cost and revenue of a wind farm to configure energy storage and determine the optimal energy storage capacity with the goal of maximizing the profit of the wind farm. In the case analysis, the effect of the deviation rate and its historical change trend, the deviation rate tolerance coefficient on electricity price is analyzed. The case analysis demonstrates the effectiveness of the proposed alterable electricity pricing mechanism and shows that the mechanism is helpful at reducing wind power output deviation and wind curtailment.

**Keywords:** alterable electricity pricing; electricity pricing mechanism; power deviation; wind power prediction; bidding of wind power; energy storage

## 1. Introduction

Due to volatility and uncertainty of wind power, actual wind output power deviates from the bidding of wind power, which has an impact on the system. The uncertainty of wind power also seriously increases the burden of the power dispatch department to control wind power [1]. The conventional power source must balance the volatility of wind output power. Present power markets are designed for trading conventional generation. For wind generation to participate in a short-term electrical market, wind power production forecasts are required. Although wind speed forecasting techniques are constantly improving, wind speed forecasts are never perfect, which will result in actual

output power of wind farm deviating from the bidding of wind power and also lead to the wind power forecasting errors that will impact on the imbalance costs for the wind farm owners. In China, wind power owners become a balance responsible player, so that the wind power owner is paying a market imbalance price for its imbalances. Different bodies have different interest appeals, the grid company pays attention to the reliability of power system, whereas the wind farm focuses on its own interests and hopes to transmit all power that wind turbines generate to the power grid, which causes the interest conflicts between grid company and wind farm.

Due to the constant changes and unpredictability of natural conditions, the actual output power of a wind farm deviates from the bidding of wind power, which often causes the problem of the insufficient regulation capability of thermal power units and the phenomenon of wind power curtailment. Experts and scholars have proposed some improved methods for mitigating the adverse effects of wind power deviation on a power system. Such methods are summarized into the following four points: (1) Improve prediction accuracy of wind output power. Literature [2–5] established a more accurate prediction model of wind power to alleviate the uncertainty of wind output power. (2) Configure energy storage to smooth the volatility characteristic of wind output power. Literature [6] presented a method to determine the energy storage capacity for wind farm to ease the balanced pressure of the power system. The method used spectrum analysis obtained by the discrete Fourier transform (DFT) of wind farm output power deviation to get the capacity of energy storage in a different time period and to adopt different control strategies to compensate for power deviation. Literature [7] pointed out that the appropriate allocation of energy storage equipment in wind farms can effectively mitigating power volatility, and the authors propose an algorithm to optimize the capacity of energy storage. (3) Ameliorate the model of source-network-load coordination and improve the consumption capacity of the power system. Literature [8] set the power supply unit, grid line, and demand response as a whole from the perspective of the power system, and designs system capacity constraints for adjusting peaking-power. The model was verified to meet the requirements of renewable energy consumption. (4) Establish an energy management system (EMS). Literature [9] pointed out that the flexible distribution of energy resources, such as energy storage and distributed generation (DG), can mitigate the randomness of DG. Literature [10,11] proposed that the energy management system or reserve capacity can increase the scheduling margin of a wind farm so that it can reliably output and reduce the deserve impact on power system. Literature [12] developed the stochastic scheduling model of the energy storing device and the thermal power station considering the indeterminacy of wind power and the load's stochastic.

All the above methods can mitigate the problem of wind power deviation on a technical level to some degree. In China, wind farm operators need to pay a compensation fee for the system dispatch department. The fee is directly proportional to the deviation of power, which has caused wind farm operators to face a near-loss income situation. In this paper, the incentive market price is considered to promote the spontaneous employment of some technical means to reduce the deviation between actual output power and the bidding of wind power. Literature [13] studies the problem of extra electric energy cost caused by the inaccurate prediction of wind power in the real-time market of Nordic countries. When the actual output of wind power is inconsistent with the bidding of wind power, wind farm operators need to pay a certain amount of extra costs to power the dispatch department according to market price of unbalanced power, but the formulation method of market price is not specified. Literature [14] studies power the market of the United States, it shows that corresponding punishment for unbalanced power will promote wind farm operators to improve the prediction accuracy and reduce output deviation, but the literature does not involve the formulation of the bidding of wind power. In this paper, we proposed an alterable electricity pricing mechanism that is specifically for China to guide wind farm operators to reduce the output power deviation, and introduced the formulation method of the mechanism in detail. What is more, the formulation method for bidding of wind power is given.

In recent years, Internet of Things (IoT) technology is developing dramatically. Literature [15] indicates that the smart grid is an important research area of the IoT. The development of the IoT will profoundly impact the transformation of renewable energy, in terms of electricity generation, scheduling, operation, and maintenance, to a more intelligent direction. After the combination of the IoT and renewable energy, the flow of information will be closely connected to that of energy, which is about to improve the ability of the information interchange. The capability of wind energy's intelligent scheduling will probably be improved so that it can contribute to the maximal consumption of quantified electricity, whereby the 'New Energy Cloud Network' is established by IoT. Because of the limit ability of the information interchange in traditional wind power dispatch, it is difficult to transmit a large amount of power generation information in real time to the dispatching control center for intelligent analysis and processing. However, under the circumstance of 'New Energy Cloud Network', the dispatching control center is able to sense the capability of wind farm's instantaneous generation, transmission of the grid's pipeline, the peak adjustment of the thermal power unit, and so forth. Moreover, the aforementioned information will be intelligently analyzed after being immediately transmitted by the cloud network, which highly probably dynamically optimize and coordinate every measure of consumption and even can achieve the consumption of renewable energy on a larger scale. Literature [16] proposes an IoT-based communication framework for the purpose of reliable communication between wind turbines and the control center.

The main work of this paper is summarized as follows: (1) an alterable electricity pricing mechanism is proposed to promote wind farms to actively reduce the deviation between actual output power and the bidding of wind power, improve the prediction accuracy, and reduce adverse impact on power system; (2) according to statistical characteristics of the normal distribution of wind power prediction error, the best bidding of wind power is proposed with the objective of least square deviation of output power. Then, an engineering application method to determine bidding of wind power is given.

## 2. Alterable Electricity Pricing Mechanism

The power dispatch department requires wind farms to provide predicted power 15 minutes in advance. The power system dispatch department organizes the real-time market according to the predicted wind power, the predicted power of the load, and the operation condition of the units. Due to the inevitable deviation of wind power, the system needs standby power to maintain power balance, which needs extra cost. Deviation between actual output power and the bidding of wind power causes the cost. Therefore, a mechanism can be designed to reduce the deviation of wind power by economic stimulus. The specific way is as follows: when there is a deviation between wind output power and predicted power, the price of the wind power grid-connection will be reduced. The larger the deviation power is, the lower price of wind power will be.

### 2.1. Selection of the Power Deviation of a Wind Farm

The objective of alterable electricity pricing mechanism is to reduce deviation rate between actual wind power and bidding of wind power, help alleviate the adverse effects of the wind power grid-connection on the power system. The basic criterion of alterable electricity pricing mechanism is related to the power deviation of wind power. The deviation rate of the wind output power is defined as the rate of the average of the difference between the actual output power and the bidding of wind power during a scheduled cycle. Deviation rate can reflect the deviation degree between output power and the bidding of wind power in each schedule cycle, which can be defined as follows:

$$D_i = \frac{1}{N_k} \sum_{k=1}^{N_k} \frac{\left| P_{\text{re},i}^k - P_{\text{dis},i} \right|}{P_{\text{dis},i}} \qquad (1)$$

where $D_i$ is the deviation rate of wind output power; $N_k$ is the number of sampling points of output power in one schedule cycle; $P_{\text{dis},i}$ is the bidding of wind power on the $i$–th schedule cycle; $P_{\text{re},i}^k$ is the actual output power of the $k - th$ sampling point in the $i - th$ schedule cycle.

## 2.2. Establishment of the Alterable Electricity Pricing Mechanism

According to the purpose of the alterable electricity pricing mechanism, two factors influencing the price of wind power are considered: the absolute value of power deviation and the trend of power deviation. The former is easy to understand. The larger the power deviation is, the more the wind farm output deviates from bidding of wind power, and the lower the feed in price will punish the wind farm. The latter mainly considers that the variation trend of wind farm deviation power can reflect its ability and enthusiasm in the regulation of power deviation, and the better its historical performance is, indicating that the stronger the regulation ability of wind farm is, the higher the price can be to reward wind farm.

The specific regulations are as follows: the grid-connection price of wind power will rise if the deviation rate is within the allowable range, that is, the wind farm in the schedule cycle is rewarded; when the deviation rate exceeds the allowable range, the grid-connection price will decrease, that is, the wind farm in this schedule cycle is punished. The grid-connection price of wind power under the alterable electricity pricing mechanism can be defined as follows:

$$\text{Price}_i = \text{p}_\text{b} \times (1 + \lambda - D_i) \times tr_i \tag{2}$$

where, $\text{Price}_i$ is real-time reward or punishment price of the $i - th$ schedule cycle; $\text{p}_\text{b}$ is reference electricity price; $\lambda$ is tolerant coefficient for deviation rate of wind output power; and $tr_i$ is variation trend of deviation rate.

## 2.3. Parameter Selection of the Alterable Electricity Pricing Mechanism

The adjustment of electricity price based on the trends of power deviation can reflect the accuracy of historical forecast power and maintain power deviation within a small range. Therefore, it is necessary to introduce an indicator that can show the trends of power deviation. Deviation rate of every schedule cycle is selected to draw a continuous curve of its changing trends. The changing trends are divided into six kinds, as shown in Figure 1. The deviation rate of type-A shows a downward trend as a whole; the deviation rate of type-B shows an overall upward trend; the curves of type-C and type-D have an inflection point, it first up and then down. The deviation rate at the end of type-C is higher than that at the beginning, whereas the type-D's is the opposite. The curves of type-E and type-F also have an inflection point, and its deviation rate first decreases and then increases. The deviation rate at the end of type-E is lower than that at the beginning, whereas the type-F's is the opposite.

The following conclusions are summarized by analyses of six types: (1) It indicated that deviation rate is decreasing when terminal deviation rate is lower than initial deviation rate, which electricity price should be increased for an ideal situation. (2) The trend of deviation rate change when there are an inflection points in curve. Therefore, the monotonicity of deviation rate curve should be considered when we calculate electricity price. The trend of deviation rate of the wind output power over the entire schedule cycle can be describe by two factors: the monotonicity of the deviation rate curve; the different values between deviation rate at the end and that at the beginning.

List deviation rate for three consecutive schedule cycles as example Terminal values and initial values of the deviation rate are known. We introduced $M_i$ to judge the monotonicity of the changing curve of the deviation rate to determine whether there is an inflection point.

$$M_i = (D_i - D_{i-1})(D_{i+1} - D_i) \tag{3}$$

Calculating the trend factor of the *i-th* schedule cycle can be divided into the following two steps.

**Step 1:** Calculate coefficient $G_i$ for the penalizing inflection point. When there is no inflection point in the changing curve of deviation rate, we record it as zero. When there is an inflection point of a changing curve, we make $G_i$ equal to the average value of $|D_i - D_{i-1}|$ and $|D_i - D_{i+1}|$.

**Step 2:** Calculate the trend factor $tr_i$.

$$G_i = \begin{cases} \frac{|D_i - D_{i-1}| + |D_i - D_{i+1}|}{2}, & M_i < 0 \\ 0, & M_i \geq 0 \end{cases} \tag{4}$$

$$tr_i = (1 + D_{i-1} - D_{i+1})(1 - G_i) \tag{5}$$

where $G_i$ is a coefficient for penalizing the appearance of the inflection point.

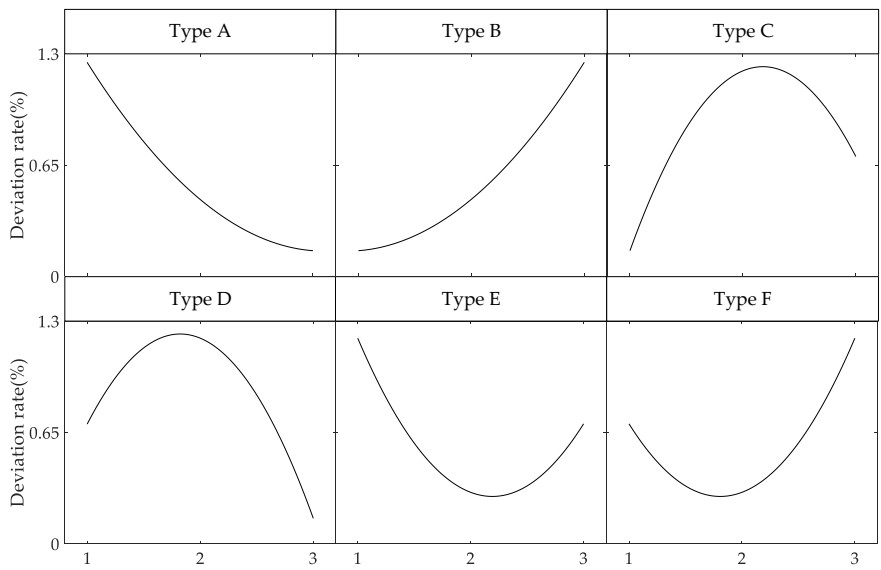

**Figure 1.** Schematic diagram of six types of changing trends.

## 3. Strategy of the Bidding of Wind Power

It can be inferred from alterable electricity pricing mechanism that the smaller the power deviation, the higher the electricity price is, and the better the economic benefit of wind farm is. This section revises wind predicted power and determines bidding of wind power.

### 3.1. Revise Wind Predicted Power

The prediction error affects the formulation of bidding of wind power. Firstly, the prediction error is analyzed. By studying prediction error data of wind output power, literature [17] found that short-term and ultra-short-term power prediction error data approximates to a normal distribution, which is defined as follows:

$$f(\frac{P_{re} - P_{pr}}{P_{cap}}) = \frac{1}{\sigma \sqrt{2\pi}} \exp((\frac{P_{re} - P_{pr}}{P_{cap}} - \mu)^2 / 2\sigma^2) \tag{6}$$

where $P_{re}$ is actual wind output power; $P_{pr}$ is wind predictive output power; $P_{cap}$ is wind farm installed capacity; $\mu$ is mean of normal distribution; $\sigma$ is standard deviation. The parameters in Equation (6) can be estimated by the method in reference [18]. The prediction error value at time $t$ is recorded as $P_{er}$. The prediction error is used to correct the prediction power.

$$P_{cor} = P_{er} + P_{pr} \tag{7}$$

where $P_{\text{cor}}$ is the revised values of predicted power.

## 3.2. Determine the Bidding of Wind Power

The deviation of power is defined as the difference between the revised predicted power and the bidding of power. The bidding of power needs to be reported to the power dispatch department, and only one bidding power value can be reported at each time interval. The deviation of output power is shown as (8). Since the deviation may be positive or negative, taking the least squares of deviation as objective function to solve trapezoidal curve of the bidding of wind power, which is defined as (9). The bidding of wind power is constant in every schedule cycle. There are $N_k$ sampling data of predictive power in a schedule cycle.

$$\varepsilon_i = P_{\text{cor},i}{}^k - P_{\text{dis},i} \tag{8}$$

$$\min \sum_{i=1}^{N_i} \left[ \frac{1}{N_k} \sum_{k=1}^{N_k} \varepsilon_i (P_{\text{cor},i}{}^k, P_{\text{dis},i})^2 \right] \tag{9}$$

where, $P_{\text{dis},i}$ is the bidding of wind power for the *i-th* scheduled cycle, $N_i$ is the total number of scheduled cycles.

For solving the above-mentioned bidding of wind power with the least squares of output power offset as an objective function, a simple derivation of binomial expansion can be used to prove that the bidding of wind power is the mean value of all predictive power in one cycle. The derivation process is as follows:

$$
\begin{aligned}
&(a_1 - x)^2 + (a_2 - x)^2 + \cdots + (a_n - x)^2 \\
&= nx^2 - 2(a_1 + a_2 + \cdots + a_n)x + (a_1{}^2 + a_2{}^2 + \cdots + a_n{}^2)
\end{aligned}
\tag{10}
$$

$$x = \frac{a_1 + a_2 + \cdots + a_n}{n} \tag{11}$$

where $x$ represents bidding of wind power, $a_1, \cdots a_n$ represents the revised predictive power.

Therefore, the bidding of wind power is the mean of predictive power sampling data in a cycle.

$$P_{\text{dis},i} = \sum_{k=1}^{N_k} P_{\text{cor},i}{}^k / N_k \tag{12}$$

where $N_k$ is the number of sampling points for predictive power in a scheduled cycle.

## 4. Case Analysis

Take a wind farm as a case to analyze the feasibility of the alterable electricity pricing mechanism. The wind farm is equipped with 10 wind turbines with 2.5MW. The electricity price is calculated according to the alterable electricity pricing mechanism proposed in this paper. The reference electricity price is 0.6 CNY/ kWh.

## 4.1. Verification of the Alterable Electricity Pricing Mechanism

We found the predicted power data and actual output power data of a wind farm in Hebei Province, China. First, we verified that the difference between above two set of data satisfies normal distribution, and calculated the revised predicted power. Utilizing the formula of (10) and (11), the bidding of power that should be solved by an optimized algorithm originally, is equivalent to the average value of the predicted power. The paper stipulates that wind farm operators predict output power every 15 minutes and reports the bidding of the power-to-power system dispatching department every hour. The bidding of power is an average of four predicted powers.

The value is shown in Figure 2. In Figure 2, the predicted power is sampled every 15 minutes, and the period of bidding of wind power is 1 hour.

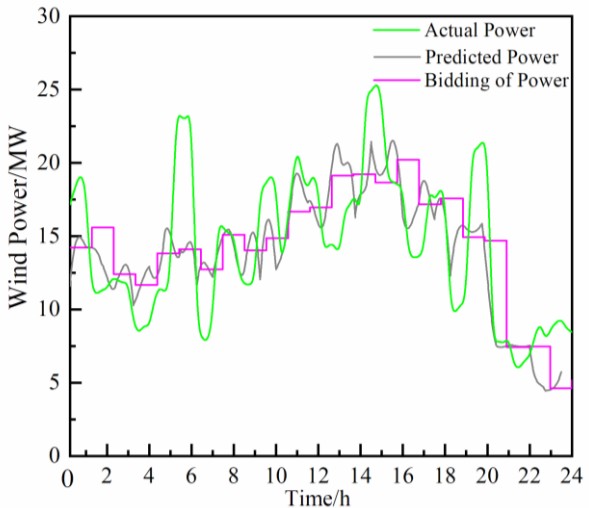

**Figure 2.** Wind power curve of a typical day

The typical daily power deviation rate $D_i$, the change trend of power deviation tri, and the electricity price Price$_i$ of the wind farm are shown in Figure 3. Comparing Figures 2 and 3, we can see that the smaller the deviation between the wind farm output and bidding of wind power, the smaller the power deviation ratio, and the higher electricity price. The smaller the power deviation trend, the higher electricity price.

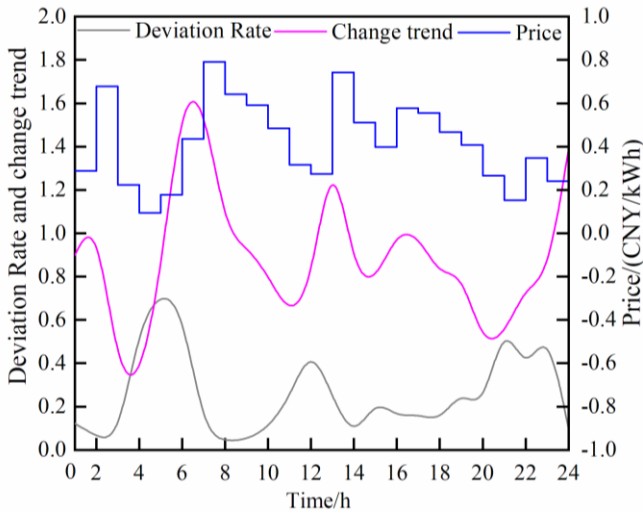

**Figure 3.** Changes in $D_i$, $tr_i$, and Price$_i$ on a typical day.

Figure 4 shows the relationship between electricity price and values of the deviation tolerance coefficient. The physical significance of $\lambda$ represents an allowable degree of a large power system for the deviation of wind output power. The deviation tolerance coefficient is directly proportional to electricity price. The smaller value of $\lambda$ means more stringent requirements on output power, which make it more difficult to obtain higher electricity price for wind farm. The values of the deviation tolerance coefficient are related to the peak regulation capability of a large system, e.g., the larger the deviation value, the greater flexibility of the large system.

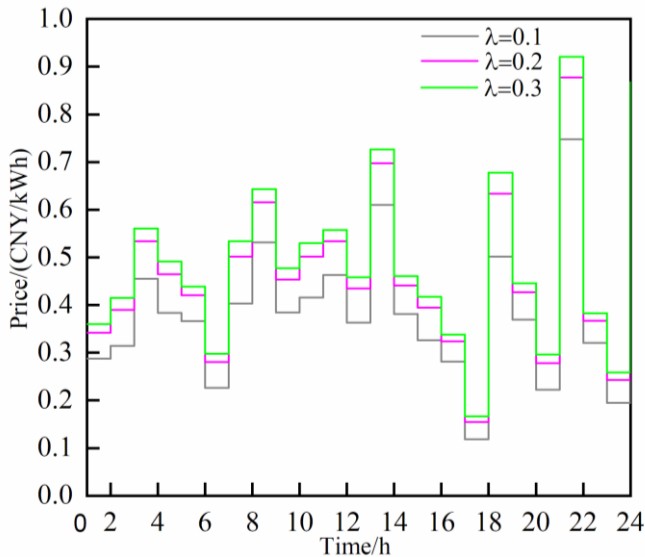

**Figure 4.** Relationship between $\lambda$ and electricity price.

### 4.2. Change of Electricity Price When Configurating Energy Storage

In the last section, the impact of $D_i$ and $tr_i$ on electricity price is verified. In the context of the implementation of the alterable electricity pricing mechanism, it needs to reduce power deviation between actual output power and the bidding of wind power if the wind farm operator wants to obtain more benefits. Energy storage can realize the migration of power and energy with time. It can effectively make up for the error of wind power prediction and reduce the deviation between the bidding of wind power and actual output power. Figure 5 shows the changes of $D_i$, $tr_i$, and Price$_i$ on a typical day after the wind farm configurates its energy storage. Compared with Figures 3 and 5, it can be seen that power deviation rate is significantly reduced after the configuration of energy storage, and the price of wind power is increased. As can be seen from Figure 6, with the increase in capacity of energy storage, the price of wind power will also increase, but the cost of energy storage will also increase apparently.

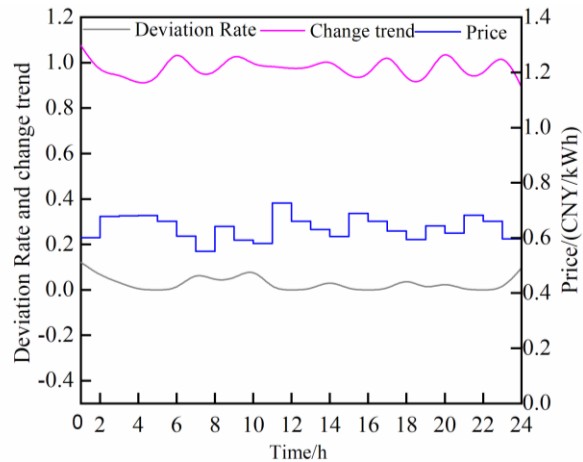

**Figure 5.** Changes in $D_i$, $tr_i$, and Price$_i$ on a typical day when the wind farm has configured its energy storage.

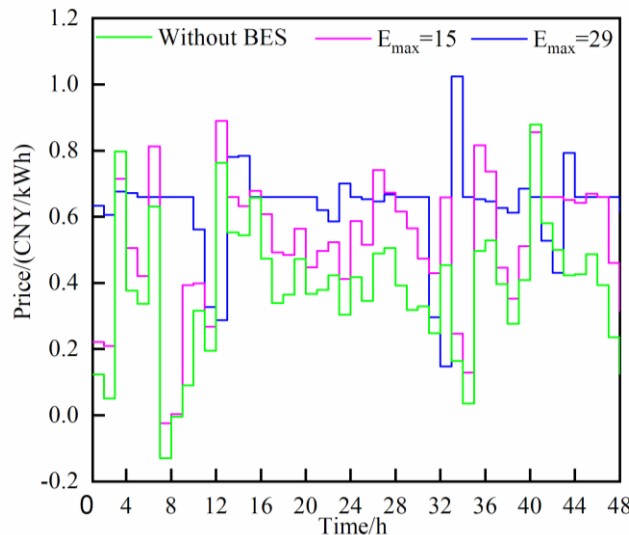

**Figure 6.** Electricity price compared with different capacity of energy storage.

Considering the cost of energy storage system (BES) and the price of wind power, the revenue expectation of wind farm is as follows:

$$\max Y = \sum \left( \text{Price}_i \times P \times T \right) - \left( C_B + C_M \right) \tag{13}$$

where $Y$ is economic profit of wind farm, the first item is the annual income obtained by wind power on-grid, and the second item is the annual comprehensive cost of BES.

The charging and discharging model of BES is shown in the Appendix A. The establishment of the charging and discharging model mainly considers the balance of power required by the wind farm and the constraints of BES charging and discharging power, so as to ensure that there will be no excessive charging and discharging power in the process of energy storage operation. The cost of BES mainly includes:

(1) Construction cost

The construction cost of battery energy storage includes the construction cost of energy-usage energy storage and power-usage energy storage, and the cost of grid-connection converters, which can be defined as follows.

$$C_B = \left( c_e E_e + c_p P_e + c_c x n_c \right) \frac{\varepsilon (1 + \varepsilon)^{L_b}}{(1 + \varepsilon)^{L_b} - 1} \tag{14}$$

where $c_e$ is the unit cost coefficient of energy-usage energy storage; $c_p$ is the unit cost coefficient of power-usage energy storage; $c_c$ is the unit cost coefficient of converters; $E_e$ is the nominal capacity of BES; $P_e$ is nominal power of BES; $x$ is a 0-1 variable, and 1 represents that energy storage system has configured converter (0 means without); $n_c$ is the number of converters; $\varepsilon$ is the discount rate; and $L_b$ is the operating life of BES.

(2) Cost of operation and maintenance

The operation cost and maintenance cost of battery energy storage include the cost of charge-discharge, the cost of power transmission, and the disposal cost of abandoned batteries.

$$C_M = \sum_{d=1}^{365} \sum_{t=1}^{24} \left( c_b^{om} (P_{d,t}^d + P_{c,t}^d) \right) + c_l^{om} P_{loss} + c_u \beta_{ES} E_e \frac{\varepsilon (1 + \varepsilon)^{Y_b}}{(1 + \varepsilon)^{Y_b} - 1} \tag{15}$$

where $c_b^{om}$ is the unit cost of charging or discharging; $c_l^{om}$ is unit loss cost of transmission; $P_{c,t}^d$ and $P_{d,t}^d$ are the charge and discharge power of BES of *t-th* time on *d-th* day, respectively; $P_{loss}$ is the loss of power during power transmission; $c_u$ is the unit-weight disposal cost of abandoned batteries; $\beta_{ES}$ is the ratio of energy to the weight of battery.

The parameters of BES are shown in [19]. When the capacity of BES is 28.9MW, the maximum annual profit of the wind farm is 101,370 CNY. Figure 7 is a comparison chart of wind power curtailment when the optimal energy storage capacity is configured in the wind farm or not. It can be seen from Figure 7 that the configuration of energy storage can greatly reduce wind power curtailment and promote the consumption of wind power.

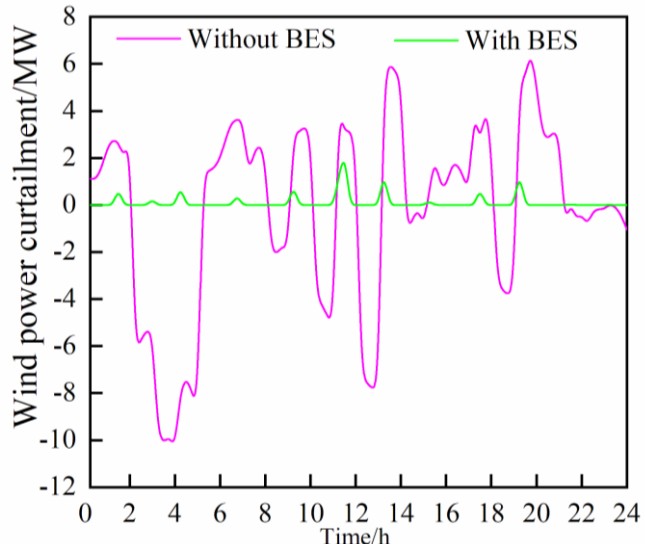

**Figure 7.** Comparison of wind power curtailment between wind farm with BES and without BES.

## 5. Conclusions

This paper innovatively puts forward an alterable electricity pricing mechanism for the purpose of mitigating the impact of auxiliary power adjustment costs on wind farm benefits caused by wind power volatility and uncertainty. An alterable electricity pricing mechanism adjusts grid-connection electricity prices according to deviation rates of wind power and its changing-trend factor. What is more, this paper determines the bidding of wind power that adopts the least square of wind power deviation as an objective function. In the context of an alterable Electricity Pricing mechanism, a wind farm operator will more proactively configure the energy storage, which will be helpful to reduce wind power deviation and promote the consumption of wind energy. In the case, we calculated that the annual profit of the wind farm is 101,370 CNY. The alterable electricity pricing mechanism provides an effective solution to the problems of wind power curtailment and wind power deviation, which will be helpful for source-network coordination in the context of marketization.

**Author Contributions:** Y.Z., Q.C. and L.E. put forward the idea of the thesis; L.E. collected the simulation data; Q.C. wrote original manuscript; Y.Z. and Q.C. reviewed and edited the article; X.Y. made funding acquisition. All authors contributed to discussing and revising the manuscript. All authors have read and agreed to the published version of the manuscript.

**Funding:** This research was supported by Open Fund of State Key Laboratory of Operation and Control of Renewable Energy & Storage System (No. NYB51201801553).

**Acknowledgments:** The authors would like to thank the editors and reviewers for their constructive comments on this study.

**Conflicts of Interest:** The authors declare no conflict of interest.

## Appendix A

When actual output power of the wind farm is higher than bidding of wind power, the remaining energy should be stored in the battery energy storage (BES) until the charged amount reaches the maximum; on the contrary, when actual output power is lower than bidding of wind power, the residual energy of BES can be used to make up for the insufficient power until the discharge amount reaches the minimum stored power of the BES, and then discharge process is stopped. Equation (A1) represents the amount of energy at *t-th* after BES execute charging process, and Equation (A2) represents the amount of remaining energy after BES is discharged. More details can be seen in [19].

$$E(t) = E(t - \Delta t) + P_{c}(t)\eta_{c}\Delta t \tag{A1}$$

$$E(t) = E(t - \Delta t) + \frac{P_{d}(t)\Delta t}{\eta_{d}} \tag{A2}$$

where $E(t)$ is BES energy at *t-th* time; $\Delta t$ is sampling interval of wind output power; $\eta_c$ is charging efficiency; $\eta_d$ is discharging efficiency; $P_c(t)$ is charging power of battery at *t-th* time, which is positive; $P_d(t)$ is the battery discharging power at *t-th* time, which is negative.

It is necessary to maintain a certain depth of charging and discharging during charging and discharging of the battery, so to constrain the depth of charge and discharge. Thus the charging power of BES at *t-th* time can be derived as follows:

$$P_{c}(t) = \min\left\{\frac{1}{\eta_{c}\Delta t}[E(t) - E(t - \Delta t)], \ \frac{1}{\eta_{c}\Delta t}[\alpha E_{\max} - E(t - \Delta t)]\right\} \tag{A3}$$

where $\alpha$ is limit coefficient of maximum charge capacity for BES, $E_{\max}$ is maximum capacity of BES.

The discharging power of BES at *t-th* time can be derived as follows.

$$P_{d}(t) = \max\left\{\frac{\eta_{d}}{\Delta t}[E(t) - E(t - \Delta t)], \ \frac{\eta_{d}}{\Delta t}[\beta E_{\min} - E(t - \Delta t)]\right\} \tag{A4}$$

where $\beta$ is limit coefficient of minimum charge capacity for BES, $E_{\min}$ is minimum capacity of BES.

Considering that charge and discharge power may affect service life of battery, the maximum charge and discharge power is regulated. Constraints for charge or discharge power of battery are as follows:

$$\begin{cases} P_{\mathrm{cmin}} \leq P_{c}(t) \leq P_{cmax} \\ P_{\mathrm{dmin}} \leq P_{d}(t) \leq P_{dmax} \end{cases} \tag{A5}$$

where, $P_{cmax}$ is the maximum value of battery charging power, $P_{\mathrm{cmin}}$ is the minimum value of battery charging power, $P_{dmax}$ is the maximum value of battery discharging power, $P_{\mathrm{dmin}}$ is the minimum value of battery discharging power.

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
