# Peer review of "Alterable Electricity Pricing Mechanism Considering the Deviation of Wind Power Prediction"

_sustainability, doi:10.3390/su12051848_

Round 1
Reviewer 1 Report
I believe I reviewed this paper before. However authors did not addressed my comments. So I am posting it again. This paper presents a reward and punishment mechanism guiding wind farms to participate in source-network coordination. Although the paper has some merits, I have few general concerns that need to be addressed carefully before I can recommend for publications:
(1) How to validate the mathematical model presented in the paper. Please explain.
(2) The motivation of this paper is somewhat vague. Please rewrite or clarify that in the introduction.
(3) Please discuss the significance of Wind farm in IoT, while discussing the following works:
i> Noor, "Sensor Fusion and State Estimation of IoT Enabled Wind Energy Conversion System". Sensors 2019
ii> J. Pan, R, "An Internet of Things Framework for Smart Energy in Buildings: Designs, Prototype, and Experiments," in IEEE Internet of Things Journal.
(4) Figures of this paper are not clear. Authors need to redraw/replot the figures for better visibility.
Reviewer 2 Report
The submitted manuscript presents the reward and punishment mechanism considering 2 deviations of wind power prediction. Though the paper highlights an important issue, I would suggest the authors proofread and revise the manuscript for better English and better clarity.
It would be better to have an alternate title of the manuscript rather than reward and punishment... Australia has a causer pay mechanism for generators so some similar titles would be more appropriate.
First two lines of abstract and introduction needs refinement.
For example:
Fluctuation and prediction errors of wind power have an impact on system. Meanwhile wind power suffers the problem of power curtailment.
This sentence must be refined for clarity. Impact on system what ?? Check for better clarity.
The introduction must present some current practices of TSOs around the world for wind farms participation in the mechanism and must discuss the proposed mechanism in comparison to them.
Section 2: Power system dispatch department requires wind farms to provide predicted power 15 minutes in advance.
Different countries have different pricing mechanism so please specify clearly that this mechanism is for a generalized system or specifically for China.
All figures must be improved for clarity. Figure numbers must be revised. fig. 1 is not mentioned anywhere in the text. In fact stated as in Fig. 2.
Section 2.3 must be improved for a better description of trends. Add more supporting lines for conclusions made in section 2.3
The manuscript must be thoroughly revised for English as it is hampering the readability and understanding. For example-Check sentence 169-170
Sentence 186- Optimize the output power of the wind farm on a typical day to get the best schedule power- Needs clarification
Overall results must be discussed with proper clarity in terms of methods for obtaining the values shown in figures.
Round 2
Reviewer 1 Report
Again authors have failed to address our comments. Their response on IoT based wind farm does not make sense. Even the topics of the paper is not IoT, the subject does have relation with the IoT. A proper discussion is required in the paper with suggested and relevant references. Also, we asked the authors to improve the figures quality. They did nothing, even they did not care to mention in the reply letter.
Author Response
In the recent years, Internet of Things (IoT) technology is developing dramatically. Reference 1 indicates that the smart grid is an important research area of the IoT. The development of IoT will profoundly impacted on the transformation of renewable energy, in terms of electricity generation, scheduling, operation and maintenance, to a more intelligent direction.
After the combination of IoT and renewable energy, the flow of information will be closely connected to that of energy, which is about to improve the ability of information interchange. The capability of wind energy’s intelligent scheduling will probably be improved so that it can contribute to the maximal consumption of quantified electricity, whereby the construction of ‘New Energy Cloud Network’ by IoT. Because of the limit ability of the information interchange in traditional wind power dispatch, it is difficult to transmit a large amount of power generation information in real time to the dispatching control center for intelligent analysis and processing. However, under the circumstance of ‘New Energy Cloud Network’, the dispatching control center is able to sense the capability of wind farm’s instantaneous generation, transmission of grid’s pipeline, peak adjustment of thermal power unit, and so forth. Moreover, the aforementioned information will be intelligently analyzed after being immediately transmitted by the cloud network, which highly probably dynamically optimize and coordinate every measures of consumption and even can achieve the consumption of renewable energy on a larger scale. Reference 2 proposes an IoT based communication framework for the purpose of reliable communication between wind turbines and control center.

Round 3
Reviewer 1 Report
Authors did not revise their paper properly. Authors need to update their paper by including their text/findings/references of the response letter in the paper. There is no point of just writing in the response letter while not reflecting those information in the paper. Please update the paper with the text and references of the response letter.
Also, I don't understand why the authors are not improving the quality of the figures? This is the 3rd time, i am raising this concern, but seems they want to ignore this point. Most of the curves are blurred, but it needs to be very visually clear.
Round 4
Reviewer 1 Report
No further comments, however, there is a lot of room to improve the quality of the figures.
This manuscript is a resubmission of an earlier submission. The following is a list of the peer review reports and author responses from that submission.
Round 1
Reviewer 1 Report
The paper is well written, the ideas are presented concisely, provides some interesting reflections about wind power dispatching policies, and it proposes a new methodology for addressing the issue of wind power participation in source-network coordination. The results obtained are conclusive and validate the proposal. Finally, the manuscript fits into the journal scope.
For all the above, the reviewer recommends the publication of the article in the present form.
Reviewer 2 Report
This paper presents a reward and punishment mechanism guiding wind farms to participate in source-network coordination. Although the paper has some merits, I have few general concerns that need to be addressed carefully before I can recommend for publications:
(1) How to validate the mathematical model presented in Section 4.3. Please explain.
(2) The motivation of this paper is somewhat vague. Please rewrite or clarify that in the introduction.
(3) Please discuss the significance of Wind farm in IoT, while discussing the following works:
i> Noor, "Sensor Fusion and State Estimation of IoT Enabled Wind Energy Conversion System". Sensors 2019
ii> J. Pan, R, "An Internet of Things Framework for Smart Energy in Buildings: Designs, Prototype, and Experiments," in IEEE Internet of Things Journal.
(4) Figures of this paper are not clear. Authors need to redraw/replot the figures for better visibility.
Reviewer 3 Report
Dear Authors,
The reviewer believes that the paper does not propose a new idea, and contribution is not strong enough for the Applied Sciences Journal.
In addition, the literature review is not supporting the idea. The paper does not include appropriate case studies.
Best Regards,
